# Innovative Mitral Valve Repair Using a Novel Automated Suturing System: Preliminary Data

**DOI:** 10.3390/medicina60071138

**Published:** 2024-07-15

**Authors:** Thomas Poschner, Severin Laengle, Sahra Tasdelen, Aldo Suria, Funda Baysal, Alfred Kocher, Martin Andreas

**Affiliations:** Department of Cardiac Surgery, Medical University of Vienna, 1090 Vienna, Austria; severin.laengle@meduniwien.ac.at (S.L.); sahra.tasdelen@meduniwien.ac.at (S.T.); aldo.suriaroldan@meduniwien.ac.at (A.S.); funda.baysal@meduniwien.ac.at (F.B.); alfred.kocher@meduniwien.ac.at (A.K.); martin.andreas@meduniwien.ac.at (M.A.)

**Keywords:** mitral valve repair, novel technology, cardiac surgery, mitral valve prolapse, minimally invasive

## Abstract

(1) *Background and Objectives:* Mitral regurgitation is a common valve disease requiring surgical repair. Even with satisfactory results, repair techniques may underlie subjectivity and variability and require long learning curves. A novel approach, the “Roman Arch” technique, may ease the technical burden. This study assessed an automated suturing device’s feasibility and time efficiency for a proposed simplified technique. (2) *Materials and Methods*: Using the MiStitch™ and MiKnot™ devices (LSI Solutions, Inc., Victor, NY, USA), the suture pattern was performed in a cadaver model. Three surgeons with different expertise levels conducted the procedures. Repair and suture placement times were recorded and analyzed. (3) *Results*: The modified “Roman Arch” repair was completed on all ten human heart specimens with an average total repair time of 3:01 ± 00:59 min and a trend toward reduced times as experience increased. The study confirmed the technical feasibility with 90% of the attempts rated as *rather satisfactory* or *very satisfactory*. (4) *Conclusions*: The MiStitch™ system effectively facilitated the modified “Roman Arch” repair in an ex vivo setting, suggesting its potential to reduce the technical complexity of mitral valve repairs. Further studies are needed to confirm its efficacy and safety in clinical practice.

## 1. Introduction

Mitral regurgitation (MR) is the second most common indication for valve surgery after aortic valve stenosis, accounting for a quarter of all valve diseases [1,2,3]. Left untreated, symptomatic MR has a mortality rate of up to 60% within five years [4,5]. The most common cause of primary MR is valve prolapse with a prevalence of 2.4% in the general population [6]. Surgical mitral valve repair (MVr) is regarded as the gold standard for the treatment of primary mitral regurgitation with an associated operative mortality rate of 1.2% [6,7,8]. 

Although standardized surgical techniques, such as those described by Carpentier, have been widely discussed, different interpretations and case numbers may lead to varying results [9,10,11,12,13,14,15]. Despite reported successful MVr rates of up to 100% (using all types of techniques), the “mere” 70% success rate in the STS database (Society of Thoracic Surgeons Adult Cardiac Surgery) highlights the need for simple and effective repair techniques [16,17,18]. As a result, Dr. Theo Kofidis has introduced a novel approach, including the “Roman Arch” technique for P2 prolapse, in which a continuous suture is placed from one papillary muscle through the prolapsing tissue to the other [19]. Although not a huge volume center, and with these techniques still extremely new, Dr. Kofidis has already clinically applied some of the proposed techniques in a few cases. The most common of those–the “Roman Arch” technique - showed auspicious early postoperative results with either no or a maximum of trace residual regurgitation. A long-term follow-up of these patients is still pending. The variability of the repair techniques should be partially reduced by implementing these techniques, instilling confidence in their reproducibility and accessibility.

With the advancement of minimally invasive surgery, novel devices may further improve mitral repair techniques. Developing and applying innovative medical devices already reduced the technical complexity with automated titanium fasteners for suture tying. Another highly interesting device, the MiStitch™ system from LSI Solutions Inc. (Rochester, NY, USA), has been extensively studied for isolated mitral valve prolapse surgery, mimicking the implantation of artificial chordae [20]. This innovative device threads an ePTFE suture through the prolapsed leaflet and the corresponding muscle, and a subsequent device secures the suture with a titanium fastener. During an ex vivo evaluation, suture placement was fast, requiring 00:55 to 08:00 min with an average of 03:36 ± 01:30 min for a single chordal replacement [21]. 

With an adjustment from the actual application, these devices could be used to simulate the “Roman Arch” technique. The objective of this study is to create a continuous suture through the prolapsing leaflet based on the proposed technique with these novel automated devices.

## 2. Materials and Methods

### 2.1. Device Technology

This study utilizes the Mi-STITCH™ Device for automated ePTFE (expanded polytetrafluoroethylene) suturing and the Mi-KNOT™ for securing ePTFE chords with a customized titanium fastener (LSI Solutions, Inc., Victor, NY, USA). These devices have been validated through extensive in vitro testing simulating clinical requirements, ex vivo cadavers, animal studies, as well as a first-in-human study. The Mi-STITCH™ Device features a handle, lever, and shaft with 360° rotation and a distal tip that articulates ~15° with an open jaw for tissue reception. The device operates by advancing curved needles through the tissue to engage and retracting the ePTFE suture through the tissue to create an untied suture loop.

### 2.2. Investigating the “Roman Arch” Repair

After thoroughly evaluating the novel technique on a plastic model, the optimal suture sequence was selected for further testing in a cadaver model, following initial trials on an ex vivo porcine heart model. These preliminary tests validated the concept and suitability of the devices for the proposed investigation. Human hearts were placed in toto and positioned to gain access to the left atrium. The left atrium was widely opened to ensure proper visualization and avoid potential interferences with limited access. Measurements such as commissural width were taken before a cleft of the P2 segment was induced by cutting several chordae tendineae. The suture pattern was as follows (see Appendix A): anterolateral papillary muscle, consecutive atraumatic leaflets bites followed by the posterior-medial papillary muscle with a subsequent suture fixation using the titanium fastener (see Figure 1). After the first leaflet bite, the suture length was carefully stabilized with forceps to determine the length of the neo-chordae.

### 2.3. Study Design

After introducing the device technology and suture pattern, three surgeons with different expertise levels—a graduate physician, a surgical resident, and a board-certified surgeon—evaluated the proposed technique. All donors had provided informed consent to donate their bodies “for the education of physicians, for continuing medical education, and for medical science”. The hearts were harvested without chemical preservation. The times and feasibility of the repairs were assessed using a comprehensive questionnaire (see Appendix A).

### 2.4. Statistics

The results of our experiments were presented using descriptive statistics. Continuous variables were expressed as means and standard deviations, while categorical variables were shown as absolute numbers and percentages. Due to the exploratory and descriptive nature of our study, sample sizes were not calculated. Differences in times between the first three and the last three experiments were calculated with the one-sided *t*-test. Statistical analysis was conducted using Microsoft Excel (Microsoft, Redmond, WA, USA).

## 3. Results

Human hearts were analyzed after liberal opening of the left atrium. Intracommissural width was an average of 36.1 ± 4.0 mm with a P2 segment of 22.5 ± 3.3 mm in width. Thanks to the cutting of 6.8 ± 2.1 tendon threads on average, a large flail was initiated in the P2 segment (with a mean width 18.7 ± 2.4 mm, proportional to the intracommissural width of 52.1 ± 6.5%).

Repair was performed in 10 specimens. The time for each experiment is depicted in Table 1. The total time for “Roman Arch” repair (defined as the first bite till the suture fixation was 03:01 ± 00:59 min (range 01:35 to 04:50 min) with an average of 02:31 ± 00:52 min (range 01:25 to 04:26 min) for the suture placement alone. In one attempt, needle pick-up was affected by a user error in the loading process, explaining the noticeable longer times of 04:50 and 04:26 min, respectively. A team learning was also evident with a strong trend to shorter overall times when comparing the first three and the last three cases (total time first three cases: 03:34 ± 00:07 min vs. 02:11 ± 00:54 min; *p* = 0.056). A total of 31 leaflet bites, with an average of 3.1 ± 0.3 bite per experiment, were performed during the testing.

Surgeons reported either *rather satisfactory* or *very satisfactory* overall feeling (90%) with only one classified as *rather unsatisfactory* due to accidental ePTFE length alteration during the deployment process as a result of a handling error in the early learning curve. The total length of the ePTFE suture correctly placed was 13.1 ± 0.8 cm.

## 4. Discussion

Following Kofidis’s proposed technique, we have successfully conducted a modified “Roman Arch” repair on ten specimens using the automated device technology [19].

While several techniques have been proposed over the decades, not all have been integrated into clinical practice [9,22,23,24,25]. In addition to various resection techniques, the “respect rather than resect” concept encourages using artificial chordae implantation to mimic native chordae tendineae and has proven results in patients with MR [26,27]. This concept may be especially intriguing for complex repairs requiring profound resections or dealing with delicate leaflet tissue or annular calcification [28,29]. The use of ePTFE chordae allows for a longer coaptation line, showing non-inferiority to the resection technique and exhibits a 90% freedom from reoperation after 18 years while also promoting faster surgical times [22,29,30,31,32]. However, challenges such as determining the correct length or accidental length alteration during the fixation process must be addressed [33,34]. Consequently, pre-measured loops are increasingly used, particularly in minimally invasive procedures, to enable valve repairs even in remote areas [35]. Nevertheless, the potential high technical complexity carries the potential risk of inadequate repair and compromised outcomes and may be more difficult in a minimally invasive setting [11,12,14,15,36].

The specific properties of ePTFE allow for cell infiltration and fibroblast attachment [37]. Vetter and Frater first demonstrated this phenomenon in the 1980s and 1990s using a sheep model with the formation of a “new tendon” around the ePTFE suture [38,39,40]. Since then, ePTFE has been used primarily for treating anterior leaflet prolapse and, subsequently also, posterior leaflet prolapse [41,42]. A particular advantage of ePTFE is that it maintains the flexibility and pliability of the suture [33,43]. The issue of potential length variations during suture fixation when knotting an ePTFE suture can be addressed using the MiKnot™ device, which provides automated fixation using a titanium fastener. Analysis of the fixed titanium fastener demonstrated the maintenance of the pull-apart forces after over 400 million cycles [44]. Consequently, using ePTFE for the modified “Roman Arch” technique seems to be a promising approach as suture material.

In patients with substantial prolapse, complex repair involving various resection techniques is sometimes necessary to achieve favorable results [9,10]. There is room for speculation based on the available data regarding the superiority of concomitant resection or isolated neo-chordae implantation. Nevertheless, isolated neo-chordae implantation may represent a less complicated form of repair [45]. Accordingly, the modified “Roman Arch” technique bears distinct potential. Not only stabilizing the prolapsing tissue but especially wrapping the posterior mitral valve leaflet (PMVL) may enable height reduction in the PMVL, thereby mitigating the risk of systolic anterior movement (SAM) [10]. Furthermore, this repair can be easily performed with only one suture, and adjusted also during the water probe, which improves flexibility during repair. Importantly, this novel technique should not be viewed as a stand-alone procedure but rather as an adaptive component of the established and proven repair techniques to gain optimal results after MVr. Moreover, using the modified “Roman Arch” technique in combination with a new adjustable annuloplasty ring could be a highly effective approach to prevent or treat recurrent mitral regurgitation [46,47].

With a range of 01:35 to 04:50 min and an average time of 03:01 ± 00:59 min, the repair in the ex vivo setting appears fast and is unlikely to represent a significant time expenditure or cause of delay. On the contrary, the support of a large P2 prolapse with two artificial chords with preformed length sutures to both papillary muscles and several times to the leaflets takes much more time compared to the technique presented herein, especially if performed minimally invasive.

Theo Kofidis describes several novel methods for a more standardized approach in the treatment of mitral valve insufficiency in two subsequent papers [19,48]. Only four of these methods have been clinically evaluated. The most common method is the “Roman Arch Pattern”, which involves suturing through the papillary muscle, the P2 segment of the free margin of the leaflet, and then the other papillary muscle. While not all techniques can be applied with MiStitch™ Technology, the modified Roman Arch method was intuitive and straightforward. Other modifications, such as the “crescent” technique (involving all segments of the PMVL) and the “longhorn” technique (starting and ending the suture pattern with the commissure), could potentially be performed using this automated suturing device. However, the impact on hemodynamics, especially the effect of “wrapping” the leaflet, would need to be separately assessed.

Integrating automated techniques holds significant promise in advancing minimally invasive procedures, as exemplified by Dr. F. Bakhtiary, who used the RAM technology (LSI Solutions, Inc., Victor, NY, USA) to establish a profound minimally invasive aortic valve replacement program [49]. Naturally, no clinical data can be generated through preclinical analysis, yet the main objective of this study—to demonstrate the feasibility of this modified technique through automated devices—was successfully achieved. Preclinical studies are crucial when introducing new techniques, with the MiStitch™ technology serving as an excellent example [20,21]. Nevertheless, one limitation of the study is the need for further in vitro testing to assess not only the feasibility but also the efficacy of this technique. Subsequently, in vivo analysis in acute animal models could generate sufficient data to enable the first human application of this proposed technique.

## 5. Limitations

The technique was only performed on an isolated (significant) posterior mitral valve leaflet prolapse. Neither the involvement of other segments nor the anterior leaflet was considered in this first ex vivo study. The repairs were carried out with ample access to evaluate the feasibility of the innovative modified repair technique. While further testing in more realistic conditions, such as the interatrial groove, may be necessary for a thorough assessment, this study’s focus was solely on evaluating a novel suture pattern using automated device technology, which was successfully achieved. Water probe testing was not performed due to inadequate ventricle filling in our experimental setup. Additionally, no further hemodynamic analyses are available at this point. This study presents preliminary data on a modified suture sequence for a simplified mitral valve repair.

## Figures and Tables

**Figure 1 medicina-60-01138-f001:**
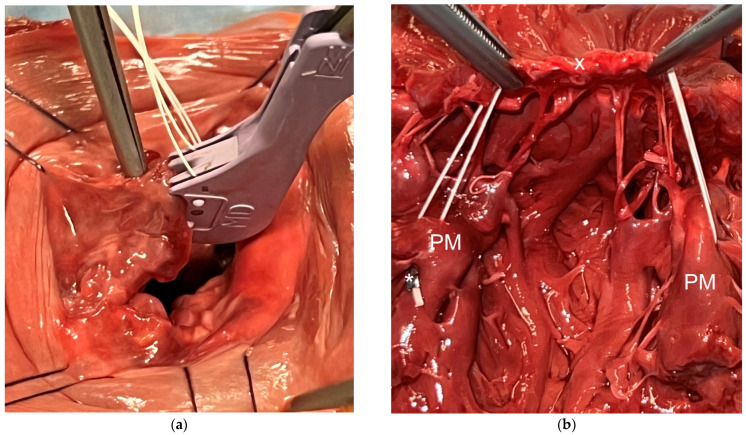
(**a**) Performing a leaflet bite using the MiStitch™ device to create the modified “Roman Arch” technique. (**b**) The final result of the repair (here the heart is cut in half through the commissure, and the anterior mitral valve leaflet is resected)—marked with “x” are the wrapping sutures of the P2 segment, marked with an asterisk (*) is the titanium fastener; PM: papillary muscle.

**Table 1 medicina-60-01138-t001:** Times measured for the “Roman Arch” repair using the MiStitch™ technology.

No	Total Time for “Roman Arch” *	Time for Suture Placement ^†^	Time for Knot Fixation ^Ω^

1	03:26	02:50	00:36
2	03:39	03:00	00:39
3	03:38	02:48	00:50
4	02:20	02:03	00:17
5	03:02	02:00	01:02
6	02:40	02:10	00:30
7	04:50	04:26	00:24
8	01:35	01:25	00:10
9	01:44	01:36	00:08
10	03:13	02:50	00:23

* defined as first papillary bite, till knot fixation [mm:ss]; ^†^ defined as first bite, till suture cut [mm:ss]; ^Ω^ defined from suture cut, till titanium fastener deployment [mm:ss].

## Data Availability

Data are contained within the article and Appendix A.

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
