# Peer review of "Innovative Mitral Valve Repair Using a Novel Automated Suturing System: Preliminary Data"

_medicina, 2024, doi:10.3390/medicina60071138_

Round 1

Reviewer 1 Report

Comments and Suggestions for Authors

One of the primary goals of cardiac surgery is to use the repair procedure, which is the gold standard in mitral valve insufficiency, in more patients and more widely. The work done to achieve excellent results in this regard is very valuable. In this context, I find this study very important in terms of its contribution to the literature.
The authors present a method in which they modified the "Roman Arch" technique, which is a new technique, using automated device technology. A more useful discussion would emerge if comments could be made as to whether this new method can be adapted to methods such as the "crescent", "closed omega" or "longhorn" technique, which are presented together with the "Roman Arch" technique.

Comments on the Quality of English Language

Minor editing of English language required

Author Response

Comment: One of the primary goals of cardiac surgery is to use the repair procedure, which is the gold standard in mitral valve insufficiency, in more patients and more widely. The work done to achieve excellent results in this regard is very valuable. In this context, I find this study very important in terms of its contribution to the literature. The authors present a method in which they modified the "Roman Arch" technique, which is a new technique, using automated device technology. A more useful discussion would emerge if comments could be made as to whether this new method can be adapted to methods such as the "crescent", "closed omega" or "longhorn" technique, which are presented together with the "Roman Arch" technique.

Answer: We greatly appreciate your positive feedback and recognition of the importance of our study in advancing mitral valve repair techniques. Your suggestions for a more comprehensive discussion on the potential adaptation of our modified "Roman Arch" technique to other existing methods such as the "crescent," "closed omega," and "longhorn" techniques are highly valuable. We have now included an expanded discussion addressing these potential adaptations in the manuscript. We believe this addition enriches the discussion and underscores the broad applicability of our method.

We believe these revisions address your concerns and enhance the robustness of our study. Thank you once again for your invaluable feedback, which has undoubtedly contributed to improving our manuscript.

Sincerely,
Thomas Poschner

Reviewer 2 Report

Comments and Suggestions for Authors

2 majors concerns : 

1. Why removing the whole left atrium (LA) and cut the heart in two. If you'd like to assess the methodology and the timing for such a surgical procedure, mi-Stitches should be used adequately through the LA and the mitral valve. I would like to see the results with this approach and not with an open heart. 

2. If the heart is kept as a whole, there would be a possibility to assess (qualitively) the repair at least with a water testing. Rapidity is not the only factor !

Otherwise intersting combination of a novel procedure with new technolgogies. 

Author Response

Comments:  2 majors concerns : 

  1. Why removing the whole left atrium (LA) and cut the heart in two. If you'd like to assess the methodology and the timing for such a surgical procedure, mi-Stitches should be used adequately through the LA and the mitral valve. I would like to see the results with this approach and not with an open heart. 
  2. If the heart is kept as a whole, there would be a possibility to assess (qualitively) the repair at least with a water testing. Rapidity is not the only factor !

Otherwise intersting combination of a novel procedure with new technolgogies. 

Answer: Thank you for your thorough review and constructive feedback. We appreciate your interest in the methodological aspects of our study and have addressed your concerns as follows:

  1. Left Atrium Resection and Heart Dissection:

We apologize for any confusion regarding the extent of the left atrium resection or the preparation of the heart. To clarify, the heart remained intact, and it was only transected for the purposes of creation of video material for proper visualization. All repairs were performed with an intact heart without cutting of the ventricle. We opted for a larger approach to assess the opportunity of the innovative modified repair technique. Although additional testing in more realistic conditions may be necessary for a thorough assessment, we focused on evaluating the suture pattern using automated device technology. To enhance clarity, we revised the set-up description in the methodology section and included the constraints of a liberal approach. We greatly appreciate your feedback about the potential misunderstanding of the set-up and hope that with the modification we have improved the understanding.

  1. Water Testing and Evaluation of Repair:

We acknowledge the importance of qualitative assessment of mitral valve repair through water testing. Due to the positioning constraints inherent in our experimental set-up, we could not properly fill the ventricle for assessment. This limitation hindered our ability to accurately evaluate coaptation and regurgitation as would be done in an operating room setting. We have included this limitation in the revised manuscript and discussed its implications. A potential follow-up study may include an in-vivo setup to test for any residual regurgitation to overcome this challenge.

We believe these comments address your concerns and enhance the robustness of our study. Thank you once again for your invaluable feedback, which has undoubtedly contributed to the improvement of our manuscript.

Sincerely,

Thomas Poschner

Round 2

Reviewer 2 Report

Comments and Suggestions for Authors

No further comments on the manuscripts. 
Thanks for the revised version which improves clarity. 

I would still have liked some kind of functional assessment instead on how fast to perform the procedure but I do understand the difficulty. One possibility would be to assess the heart on an ex vivo bio simulator.